# Amniotic Fluid Cells, Stem Cells, and p53: Can We Stereotype p53 Functions?

**DOI:** 10.3390/ijms20092236

**Published:** 2019-05-07

**Authors:** Melissa Rodrigues, Christine Blattner, Liborio Stuppia

**Affiliations:** 1Department of Psychological, Health and Territorial Sciences, Laboratory of Molecular Genetics, School of Medicine and Health Sciences, G. d’ Annunzio University, Chieti-Pescara, 66013 Chieti, Italy; melissarodrigues14@gmail.com; 2Institute of Toxicology and Genetics, Karlsruhe Institute of Technology, P.O. Box 3640, 76021 Karlsruhe, Germany; 3Centre of Aging Science and Translational Medicine (Ce.S.I.-Me.T.), G. d’Annunzio University, Chieti-Pescara, 66013 Chieti, Italy

**Keywords:** amniotic fluid cells, embryonic stem cells, mesenchymal stem cells, p53, proliferation, differentiation

## Abstract

In recent years, great interest has been devoted to finding alternative sources for human stem cells which can be easily isolated, ideally without raising ethical objections. These stem cells should furthermore have a high proliferation rate and the ability to differentiate into all three germ layers. Amniotic fluid, ordinarily discarded as medical waste, is potentially such a novel source of stem cells, and these amniotic fluid derived stem cells are currently gaining a lot of attention. However, further information will be required about the properties of these cells before they can be used for therapeutic purposes. For example, the risk of tumor formation after cell transplantation needs to be explored. The tumor suppressor protein p53, well known for its activity in controlling Cell Prolif.eration and cell death in differentiated cells, has more recently been found to be also active in amniotic fluid stem cells. In this review, we summarize the major findings about human amniotic fluid stem cells since their discovery, followed by a brief overview of the important role played by p53 in embryonic and adult stem cells. In addition, we explore what is known about p53 in amniotic fluid stem cells to date, and emphasize the need to investigate its role, particularly in the context of cell tumorigenicity.

## 1. Introduction

Stem cells are present throughout the embryonic development and in adult organs. A coordinated control of stem cell self-renewal and differentiation is essential for maintaining tissue and organ homeostasis. Since 1980, a huge amount of work has been invested into the analysis of embryonic stem (ES) cells. These cells show incredible features like i) the ability to be expanded in vitro while maintaining an undifferentiated state, and ii) pluripotency, which means that the cells possess the capability to differentiate into every cell type of the adult body [1]. Because of these properties, ES cells may potentially be used in regenerative medicine in the future. However, the use of ES cells in stem cell therapy is also raising ethical concerns since they are isolated from human embryos, and they possess the risk of tumor formation in vivo. Pertaining to this issue, Yamanaka and Takahashi showed that stem cells with properties similar to ES cells can be generated from mouse fibroblasts by simply introducing combinations of genes like *Oct4*, *Nanog*, *c-Myc*, and *Klf4* [2]. They designated these cells as “induced pluripotent stem cells” or “iPS” cells. Although iPS cells have clinical potential as a source of cells for regenerative medicine similar to ES cells, transplanting differentiated cells derived from iPS cells into patients remains a grave concern, as the genomic integrity of these cells and the safety of the patient is still an issue [3]. A second problem is the low efficiency and slow kinetics of iPS cell generation in vitro [3]. To overcome these limitations, researchers started to look for alternative sources of stem cells. This endeavor gave rise to research in the field of perinatal stem cells. 

Perinatal stem cells can be derived from postembryonic tissues, which include the tissues sourced at the time of birth, but also comprise the time period from the 16th week of gestation through the neonatal period [4,5]. These tissues include the amniotic fluid, the placenta, placental membranes (amnion, chorion and Wharton jelly) and umbilical cord [6,7,8,9,10]. At the time of birth, these tissues are usually discarded as “biological waste”. As these tissues are anyway discarded, harvesting stem cells from these sources is a simple and non-invasive method for obtaining stem cells that could be used for therapy. Interest in perinatal stem cells was particularly initiated, when Kaviani and colleagues reported in 2001 about the use of these cells for tissue engineering and for the surgical repair of congenital anomalies in the perinatal period [11]. In addition to being easily accessible, perinatal stem cells can be isolated, expanded, and differentiated in vitro [12,13,14,15,16,17]. It is therefore anticipated that these cells can serve as a novel source and an alternative to human ES cells for research and therapy.

The amnion encloses the amniotic cavity containing the amniotic fluid, a protective and nutrient-containing liquid for the developing fetus [18]. It is mainly composed of water, electrolytes, chemical substances, nutrients, and cells shed from the growing embryo [19,20]. Among the heterogeneous population of amniotic fluid cells, a class of multipotent cells, the amniotic fluid stem (AFS) cells have been identified. These cells share characteristics of ES and adult stem cells [21]. Most interestingly, and in contrast to ES cells, the AFS cells are not tumorigenic when injected into immune-compromised animals [14,22]. This property makes these cells particularly attractive for clinicians and researchers in the field of regenerative medicine. A comparison between the main features of ES and AFS cells is shown in Table 1. 

By using various factors and methods, AFS cells can also be reprogrammed into iPS cells [23,24,25]. Human AFS-iPS cells express pluripotency markers such as alkaline phosphatase, Oct4, Sox2, SSEA4, etc., and can be continuously propagated in vitro while maintaining their normal karyotype [26]. As AFS cells have a stem cell-like characteristic, ectopic expression of the transcription factor Oct4 is sufficient to make them fully pluripotent, at least under certain conditions [27]. In a recent study, AFS cells from human second trimester amniotic fluid were also transfected with modified messenger RNAs of *Oct4*, *Klf4*, *Sox2*, *Lin28,* and *c-Myc* to induce a pluripotent state, and then differentiated into functional cardiomyocytes using inhibitors of glycogen synthase kinase 3 (GSK3) and Wnt [25]. Cells from the first trimester that have been selected for the surface antigen c-kit can furthermore be fully reprogrammed to pluripotency without transfecting ectopic factors when they are cultured on matrigel in cell culture medium that has been supplemented with the histone deacetylase inhibitor, valproic acid [28]. 

The lack of tumorigenesis after transplantation is an interesting feature of AFS cells, although no information is available regarding the underlying mechanisms. Important clues could be gathered by investigating in AFS cells the activities and functions of crucial cell cycle regulators, like the tumor suppressor gene p53.

p53 is one of the most well-known and most intensively investigated tumor suppressor proteins. A lot of work has already been done on investigating the role of p53 in ES cells and other adult stem cells and it has been established that apart from its traditional tumor suppressor function, p53 is also reported to be involved in controlling Cell Prolif.eration, self-renewal, and differentiation of stem cells [29]. 

## 2. Phenotypic Characterization of Amniotic Fluid Stem Cells

Amniocentesis is routinely performed between 16–18 weeks of pregnancy. The collected amniotic fluid is used for prenatal genetic testing and as a source of AFS cells. Different approaches and protocols have been used to isolate AFS cells [8,30,31]. These isolation protocols can be distinguished into i) a one-step culture protocol; in this procedure the primary amniocyte culture is left undisturbed for seven [32,33] to twenty [34] days without any changes in the medium; ii) a two-step culture protocol; in this approach the non-adherent amniocytes are collected on the fifth or seventh day after isolation and kept until further analysis is completed. After that, the cells are cultured in the appropriate medium [31,35]; iii) a procedure where the amniocytes are selected for a specific marker, for example for CD117 (c-Kit receptor) [8,36] and iv) a short-term culture method; specimens are centrifuged and incubated in Mesencult or Chung medium. After three days, non-adherent cells are discarded, and fresh medium is added. Thereafter cells can be cultured [37] or clonally expanded [38] (Figure 1). 

Based on morphological and growth characteristics, human amniotic fluid cells are grouped into three main categories: i) amniocytes or amniotic fluid specific AF-type cells; this type of cell comprises about 60% of the amniotic fluid cells. ii) Epitheloid E-type cells; this type of cells make about 34% of the amniotic fluid cells. iii) Fibroblastic F-type cells; about 6% of the amniotic fluid cells belong to this type of cells [39,40,41]. The first and second group of cells are usually present at the beginning of cultivation. The third group of cells appears later, when the cells have been cultured already for some time, while the E-type cells disappear when the amniotic fluid cells are taken into culture [41,42,43]. The majority of the AFS cells share a multipotent mesenchymal phenotype similar to adult mesenchymal stem cells [8,31,34,44,45,46,47,48]. In consistency with their multipotent phenotype, AFS cells express the pluripotency markers Oct4, SSEA-4, and Nanog at the RNA and protein levels, as well as the stem cell markers vimentin and alkaline phosphatase [49,50]. Other stem cell markers like SSEA-3 and TRA-1-81 are, however, usually not present in AFS cells, indicating that these cells can most likely not be regarded as fully undifferentiated cells [8,28]. Interestingly, cells isolated from some amniotic fluid cultures express neural markers, such as Nestin, βIII-tubulin, GFAP and NEFH [22]. 

Approximately 1% of the AFS cells express the surface antigen c-kit (CD117), a receptor tyrosine kinase type III, which plays a role in cell survival, proliferation, and differentiation [8]. Some authors regard only the c-kit positive cells as the stem cell population in a heterogeneous AFS cell culture [8]. However, both the c-kit-positive and the c-kit-negative amniotic fluid cells have similar morphology and proliferation characteristics and are able to differentiate along the osteogenic and adipogenic lineages under specific culture conditions. The c-kit-positive cells, however, show high Oct-4, Sox2, and Nanog expression and also show a higher ability to differentiate into myocardial-like cells than c-kit-negative cells, indicating that the c-kit-positive cells may have more stem cell properties than the c-kit-negative cells [17]. The fibroblast-shaped cell population of amniotic fluid cells furthermore expresses the mesenchymal markers CD90, CD105, CD73, and CD166, while the cells are negative for the hematopoietic markers CD45, CD34, and CD14 [14]. Based on the fact that these cells express some but not all stem cell markers and several mesenchymal markers, some researchers consider AFS cells to be mesenchymal stem cells [14]. 

AFS cells maintain a normal karyotype and stable telomeres when taken into culture, even after over 200 population doublings [49], yet other properties like expression of stem cell pluripotency markers, expression of specific proteins during differentiation, immune regulation and endothelial regenerative potential after differentiation may vary with gestational stage [51,52,53].

One of the most important qualities of AFS cells is that they are non-tumorigenic when transplanted into immunocompromised mice [8]. Although AFS cells possess several similarities with ES cells (Table 1) and the transcriptome of AFS cells from the first trimester with passage numbers between 15 and 20 shows about 80% identity with that of human ES cells [54], AFS cells differ from ES cells in this crucial feature. Even seven months after the intravenous injection of AFS cells into mice, the animals had no detectable tumors [55]. Although many groups have evidenced the non-tumorigenic property of these cells, detailed investigations on this inability have not been reported so far. Tumorigenicity is usually associated with chromosomal alterations, and since clonal human AFS cell lines lack chromosomal alterations, it is likely that their chromosomal stability is responsible or at least contributes to the fact that these cells do not form tumors. However, when iPS cells are generated by ectopic expression of Oct4 in AFS cells, tumors are formed within 2 months [27].

Likewise, AFS cells also do not form teratomas after in vivo transplantation [22]. Even 11 weeks after transplantation of cultured human AFS cells into immunodeficient mice, teratomas were not observed while under the same conditions human ES cells caused teratoma formation after only 3–4 weeks [22]. On careful consideration, teratoma formation is closely linked with pluripotency. However, AFS cells are not pluripotent, and it is most likely that the fact that AFS cells are not pluripotent makes teratoma formation impossible. This idea is supported by the observation that certain in vitro culture conditions equip AFS cells with the ability to form teratomas. This ability of AFS cells to form teratomas comes along with an upregulation of pluripotency associated genes like *Oct4*, *Nanog,* and *Sox2* [28]. 

## 3. Differentiation Potential of Amniotic Fluid Stem Cells

Since the discovery of AFS cells, several groups have been exploring their differentiation potential. Even before it was known that the amniotic fluid contains Oct4-positive stem cell-like cells, Streubel and coworkers had demonstrated the expression of skeletal muscle proteins when amniotic fluid cells were cultured in the supernatant of rhabdomyosarcoma cell lines [56]. After this first observation of a differentiation potential of AFS cells, in the presence of various inducing factors, these cells were reported to be able to differentiate into multiple lineages including hematopoietic, neurogenic, osteogenic, chondrogenic, adipogenic, renal and hepatic lineages and to form fibroblasts, adipocytes, and osteocytes (Figure 2) [8,35,36,44,45,57,58,59,60,61,62,63,64]. 

Guan and colleagues showed that undifferentiated human AFS cells express several cardiac genes (transcription factor mef2, the gap junction connexin43 and H- and N-cadherin), acquire a cardiomyocyte-like phenotype after 24-h incubation with 5-aza-2′-deoxycytidine (5-AZA-dC) and develop mechanical and electrical connections with neonatal rat cardiomyocytes when co-cultured [65]. Rat AFS cells were induced towards cardiomyogenic differentiation by co-culturing with neonatal rat cardiomyocytes [66] and under specific conditions mouse AFS cells were shown to differentiate into cardiomyocyte-like beating cells, although with a very low efficiency [67]. More recently it was demonstrated that a subset of amniotic fluid cells that were positive for CD90, called cardiopoietic AF cells, might differentiate toward the cardiac lineage giving rise to cardiomyocyte-like cells [68]. 

AFS cells have furthermore been demonstrated to form embryoid bodies when cultured in suspension, accompanied by a decrease in the expression of the pluripotency stem cell marker Oct4 and nodal, a marker for the self-renewal potential of ES cells, followed by the induction of differentiation markers like Flk1 (fetal liver kinase 1) for endothelial cells, E-cadherin for epithelial cells, Pax 6 (paired box protein 6) for ectodermal cells, TBXT (T-box transcription factor T) and HBE1 (hemoglobin subunit epsilon 1) for mesodermal cells, and GATA4 for endodermal cells [69]. 

Similar to ES cells and iPS cells, AFS cells are capable of differentiating into primordial germ cells when cultured under appropriate culture conditions and through formation of embryoid bodies [70,71]. Thus, AFS cells may even be used to study the mechanisms of early gametogenesis.

In summary, the enormous efforts in the past years by various groups to exploit and investigate differentiation of AFS cells have yielded positive results and demonstrate that amniotic fluid cells have a broad differentiation potential.

## 4. p53, the Guardian of the Genome

Functionally, p53 is a transcription factor that elicits its cellular activity mostly through transcriptional activation of target genes. However, besides its primary function as a transcription factor, p53 can also promote apoptosis independent of transcription via direct interaction with pro- and anti-apoptotic proteins [72]. Pertaining to its anti-proliferative activity, p53 is under tight control. The activity of p53 is regulated by its protein abundance as well as by its post-translational modifications. It is well established that p53 degradation is largely achieved through the ubiquitin-proteasome pathway [73], but a role for other proteolytic enzymes such as calpain has also been reported [74]. The short half-life of p53 ensures that its abundance in non-stressed cells is low. However, when its activity is required, p53 is protected from degradation and accumulates to high levels [75,76]. The major regulator of p53 levels is the oncoprotein Mdm2, to which p53 is connected by a negative feedback loop. The p53 protein binds to the *mdm2* gene and activates its transcription, and the resultant Mdm2 binds to p53, blocks its activity, and mediates the proteolytic degradation of p53 [77]. As a main negative regulator of p53, Mdm2 is fine-tuned by a group of protein partners that bind to Mdm2 in a context- and cell- dependent manner. One of these proteins is Mdm4 (Mouse double minute 4 or MdmX) [78], an Mdm2 homolog which oligomerizes with Mdm2 and increases Mdm2’s E3 ligase activity [79]. This Mdm2/Mdm4 oligomerization allows Mdm2 to more efficiently regulate p53 levels and activity [80]. Additionally, apart from Mdm2 and Mdm4 other E3 ubiquitin ligases like constitutive photomorphogenesis protein (COP1) or p53-induced protein with a RING-H2 domain (Pirh2) regulate protein levels and activities of p53 in a variety of physiological conditions, and in response to cellular changes induced by stress [81]. COP1 and Pirh2 synergistically function with the Mdm2/Mdm4 complex to inhibit p53’s activities, including p53’s transcriptional activity [80].

p53 is part of a multiple gene family that also includes p63 and p73, as well as several splice variants of the three proteins [82,83]. p53 splice variants were first identified in 1980s in humans and mice [84,85]. Although p53, p63 and p73 share structural, biochemical and biological homologies and regulate a common set of target genes, they control different cellular activities. For instance, the p53 protein has a fundamental role in growth control and maintenance of genomic integrity, whereas p63 is essential for epidermal morphogenesis and limb development [86]. On the other hand, mice that are deficient in p73 have abnormalities of the nervous system and suffer from chronic infections and inflammation [86]. 

The role and effector functions of p53 however are context dependent and such context dependence is affected by cell type, genetic background of the cell, cell microenvironment, and nature of stress the cell is undergoing. Because p53 is such a critical cellular protein, multiple mechanisms have evolved to modulate its activity. The existing strategies consist of manipulating p53’s transcriptional or translational control by altering the level of p53 mRNA, by changing its posttranslational modifications, by altering p53’s half-life by freeing wild type p53 from MDM2, and by altering p53’s intracellular localization [87]. 

## 5. p53 in Stem Cells

Similar to differentiated cells, p53 becomes activated in stem cells in response to DNA damage, where p53 activates the promoters of its canonical target genes including *Bax* and *Puma* resulting in apoptosis [88]. In addition, p53 suppresses the expression of *Nanog* and *Oct4* after DNA damage, thus evoking differentiation of these cells [89,90,91]. Both activities contribute to the maintenance of genomic stability in stem cells. 

However, p53 is not only active in stem cells after DNA damage. A few years ago, it was discovered that p53 is also active in non-stressed stem cells. Yet, the anti-proliferative activity of p53 is compromised in these cells and p53 instead directs a transcriptional program that is highly reminiscent to that of tumor-derived mutant p53 [92]. 

More recent studies implicate p53 also into the plasticity of stem cells. Here, p53 seems to be involved in the regulation of the self-renewal of stem cells as well as for the onset of differentiation (Figure 3). The emerging non-canonical roles of p53 in stem cells have been nicely summarized earlier [29] and were also reviewed very recently by Olivos and Mayo [93]. 

The first evidence for a role of p53 in embryos dates back to the 1980’s, when this protein was found highly expressed in primary cell cultures obtained from 12–14 day old mouse embryos but not in 16-day-old mouse embryos [94]. A year later it was reported that the amount of p53 protein decreased significantly during embryogenesis [95]. This observation was further supported by many other groups who also observed a significant decrease in the amount of p53 mRNA and protein during differentiation [90,96,97,98]. Interestingly, Sabapathy et al. observed a conformational change in p53 during differentiation resulting in loss of functional activity that allows the differentiating cells to escape from apoptosis [97]. Lin and coworkers reported that in murine ES cells, p53 directly represses the stem cell marker Nanog [90], a protein known to promote dedifferentiation of astrocytes into cancer stem-like cells [99]. Subsequently, it was shown that spontaneous differentiation of human ES cells is significantly reduced when p53 abundance is decreased and that disrupting or inactivating p53 increases the production of iPS cells, a feature of p53 that is conserved from mice to men [100,101,102]. Few years later, another group found out that p53 actively promotes differentiation of ES cells by inducing the expression of *miR-34a* and *miR-145*. The latter one operates as a direct repressor of the pluripotency factors Oct4, Klf4, Lin28A, and Sox2 upon retinoic acid-mediated differentiation of ES cells [103]. Likewise, treatment of human ES cells with Nutlin, a small molecule that binds to the p53 antagonist Mdm2 and inhibits p53 degradation resulting in p53 accumulation promoted the differentiation of stem cells [104]. Maimets and coworkers assumed that the main mechanism that is responsible for the differentiation of human ES cells after activation of p53 is the inhibition of S phase entry into the cell cycle due to the induction of the cell cycle inhibitor p21 by p53. Other studies showcasing p53^−/−^ human ES cells further confirmed the suppressive activity of p53 for stem cell pluripotency [102,105,106]. Of note, silencing Oct4 in human ES cells led to the activation of p53 and in consequence to ES cell differentiation [107]. Silencing of Oct4, furthermore, reduced the expression of Sirt1, a deacetylase known to inhibit p53 activity, leading to increased acetylation of p53 at lysine 120 and 164 which leads to stabilization and activation of p53 in human ES cells [107]. A very recent report identified a regulatory network in mesendodermal differentiation of mouse and human ES cells involving other members of the p53 family (p63 and p73) [108]. They showed p53, p63, and p73 are crucial for the activation of eomesodermin, Foxa2 (forkhead box A2) and goosecoid which support the differentiation into mesendoderm of ES cells in vitro and in the embryo. p53 family members directly control the activation of these genes by governing the cooperation of Wnt and nodal which are interdependent in driving mesendodermal differentiation of pluripotent cells [108]. 

In adult stem cells including neural and hematopoietic stem cells, p53 negatively regulates proliferation and self-renewal and helps to maintain the quiescent state [109,110]. More recently, it was uncovered that p53 loss had a pro-osteogenic function in primary mouse bone marrow stem cells, where p53 indirectly repressed the expression of Runx2 (runt related transcription factor 2) by activating the *miRNA-34* family [111]. Furthermore, this lineage specification role for p53 appeared to be conserved in human osteosarcoma cells [111]. 

## 6. p53 in Amniotic Fluid Stem Cells

While the role of p53 has been investigated in ES cells and adult stem cells as described above, only little was known about the regulation and function of p53 in AFS cells in the past. Recently, our group investigated p53 protein levels, p53 localization, and p53 activity during Cell Prolif.eration, differentiation, and during the DNA damage response in human AFS cells [112]. 

Earlier reports showed both constant stable expression and also downregulation of p53 mRNA during long term passaging of AFS cells and AF mesenchymal stromal/stem cells [14,113,114]. In our hands, expression of p53 did not change with increased passaging numbers of AFS cells, which is consistent with the studies of Poloni et al. and Phermthai et al. [14,113]. In contrast to our study, Savickiene et al., reported increased expression of p53 and its target gene p21 during the propagation of AFS cells from passages 3–9 [115]. This increase in p53 levels was associated with cellular senescence as assessed by monitoring the β-galactosidase activity while we did not observe an increase in cellular senescence with increased propagation of our AFS cells. The reason for these contradictory results is, however, unknown. We furthermore found that undifferentiated AFS cells express p53 at lower levels than in cancer cells and that the p53 protein in AFS cells is mainly localized in the nucleus. However, despite its nuclear localization, the anti-proliferative activity of p53 is compromised in these cells as it has been seen previously in ES cells [92,112]. 

When we treated AFS cells with retinoic acid to induce neural differentiation, we observed an increase in p53 protein levels [112] which is consistent with the studies of Thangnipon et al. and Gasiūnienė et al. Thangnipon and his crew observed increased p53 levels when they induced neuronal differentiation of AFS cells with a combination of N-benzylcinnamide and bone morphogenetic protein (BMP)-9 [116] whereas Gasiūnienė and colleagues observed an increase in p53 mRNA and protein levels upon cardiomyogenic differentiation in AFS cells using DNMT methyltransferases and p53 inhibitors [117]. We could furthermore show that p53 is an important determinant for the DNA damage response of AFS cells. Upon DNA damage, p53 becomes activated in AFS cells and leads to the transcriptional activation of its target genes *p21* and *mdm2* [112]. In addition, nuclear staining for the DNA damage marker γ-H2AX was negative in normal AFS cells in comparison to cells whose DNA was damaged with etoposide [112].

## 7. Concluding Remarks

AFS cells are emerging as an alternative source of stem cells with promising advantages for disease modeling and applications in regenerative medicine, although there are still questions to address regarding their safety and potential, and p53 may play important roles in these processes. As the amniotic fluid contains a heterogeneous population of cells, in many studies, mixtures of unselected, non-clonal cells have been used, causing conflicting results and uncertainty regarding the identity of the cell population employed. However, stem cells within the amniotic fluid do possess a high proliferation rate, clonogenicity, and a broad differentiation potential ranging from multipotent cells to unipotent committed progenitors. The therapeutic potential of AFS cells has been shown in experimental models of diseases of the lung, heart, skeletal muscle, bone, cartilage, kidney, blood and nervous system. Since AFS cells are primary cells of a very early stage of human development, these cells are also interesting as a model system for basic research. Finally, although AFS cells have profiles similar to ES and adult stem cells, it is quite surprising why these cells are not tumorigenic when transplanted into immune-compromised animals. p53, which is already known to be an important tumor suppressor protein in pluripotent stem cells and differentiated cells, may be considered as a crucial control point in AFS cells as well. Therefore, it is essential to identify if p53 is involved not only in the processes underlying the ‘stemness’ of AFS cells but also in their tumorigenicity.

p53’s role in tumorigenesis is widely known with thousands of publications and an increasing list of p53 targets and effector functions, making it even more difficult to understand the complexity and reach of this single protein. Interest is also driven to focus on potential mechanistic links between the loss of p53 and the stem cell–like cellular plasticity which has been suggested to contribute to tumor cell heterogeneity and to drive tumor progression. We now know that p53 is actively present and involved in a variety of cellular processes in AFS cells, but its involvement is less well understood. Does p53 play a different cellular function in AFS cells? Is p53 essential for tumor suppressive functions in AFS cells? Does the loss of p53 in AFS cells lead to tumor formation? These are yet unanswered questions, the knowledge of which could potentially improve the usage of AFS cells for therapy in the future. 

## Figures and Tables

**Figure 1 ijms-20-02236-f001:**
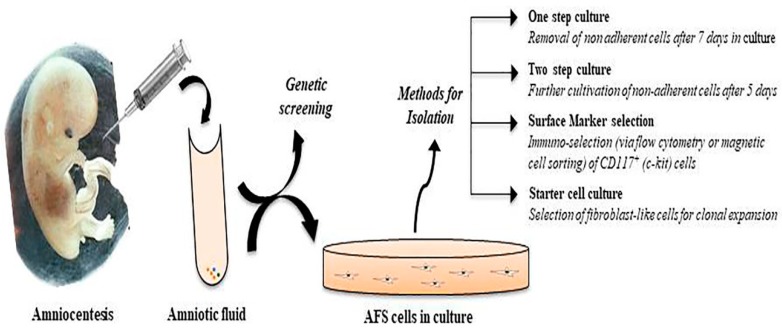
Isolation of human AFS cells and establishment of cell cultures.

**Figure 2 ijms-20-02236-f002:**
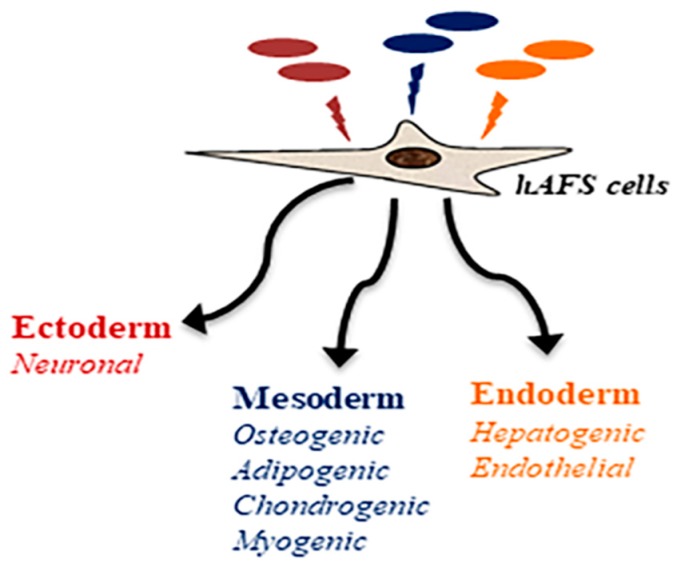
Multipotent differentiation potential of AFS cells.

**Figure 3 ijms-20-02236-f003:**
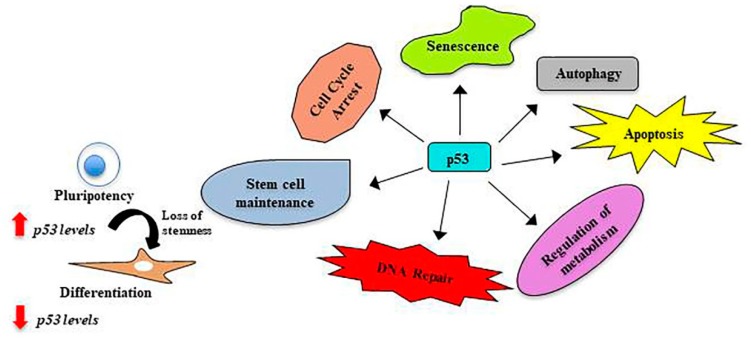
Diverse roles of p53 in stem cells.

**Table 1 ijms-20-02236-t001:** Comparison between human embryonic stem (ES) cells and human amniotic fluid stem (AFS) cells.

Features	Human ES Cells	Human AFS Cells
Source of cells	Inner cell mass of a blastocyst stage embryo	Amniotic fluid from second or third trimester of pregnancy
Plasticity	Pluripotent	Multipotent
Ease of cultivation	Requirement of MEF for feeders and the presence of bFGF	Cultivation without feeders but in presence of bFGF
Doubling time	24 to 96 h	Approximate 36 h
Marker expression: *Pluripotency*	Oct4, Nanog, Sox-2, Klf4	Oct4, Nanog (low), Sox2 (low), Klf4 (low)
*Cell Surface Antigens* *Mesenchymal*	SSEA1, SSEA3, SSEA4, Tra-1-60, Tra-1-81	SSEA4, CD-117 (c-kit), CD90, CD105, CD73, CD13, CD29, CD44, CD146
In vitro differentiation potential	All three germ layers (under specific culture conditions)	All three germ layers (under specific culture conditions)
Ability to form embryoid body	Yes	Yes
Tumorigenic in vivo	Yes	No
Ethical concerns	Yes	No
Legal restrictions	Yes	No
Therapeutic animal model testing	Yes	Yes

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
