# Peer review of "Amniotic Fluid Cells, Stem Cells, and p53: Can We Stereotype p53 Functions?"

_ijms, 2019, doi:10.3390/ijms20092236_

Round 1
Reviewer 1 Report
This timely review summarizes recent findings related to the use of Amniotic Fluid Stem (AFS) Cells in regenerative medicine and and the role of the p53 tumor suppressor gene in maintenance of genom stability and prevention of tumorigenesis of these cells. The review is well and clearly written, I enjoyed in reading it.It is in a good form to be publsihed.
I have a very minor point. In the review it is reported that AFS cells show no chromosomal alterations, and indeed they do not form tumors when injected in mice. This is not the case of embryonic stem cells and inducued pluripotent stem cells that show chromosomal abnormalities and DNA damage. Hence, I was wondering whether anyone has ever analyzed if ASF cells display DNA damage, by checking for nuclear staining of DNA damage markers such as gH2AX, 53BP1, RPA and RAD51 for instance. Also, is it know what the cell cycle of these cells look like? Is it simlilar or different from that of ES and.or iPSs?
Author Response
Response to Reviewer 1 Comments
Comment 1: This timely review summarizes recent findings related to the use of Amniotic Fluid Stem (AFS) Cells in regenerative medicine and the role of the p53 tumor suppressor gene in maintenance of genome stability and prevention of tumorigenesis of these cells. The review is well and clearly written, I enjoyed in reading it. It is in a good form to be published.
I have a very minor point. In the review it is reported that AFS cells show no chromosomal alterations, and indeed they do not form tumors when injected in mice. This is not the case of embryonic stem cells and induced pluripotent stem cells that show chromosomal abnormalities and DNA damage. Hence, I was wondering whether anyone has ever analyzed if ASF cells display DNA damage, by checking for nuclear staining of DNA damage markers such as gH2AX, 53BP1, RPA and RAD51 for instance. Also, is it know what the cell cycle of these cells look like? Is it similar or different from that of ES and or iPSs?
Response 1: Thank you for your valuable comments. Regarding your question if AFS cells display DNA damage, we observed γ-H2AX staining was negative in normal AFS cells in comparison to the cells where DNA damage was induced with etoposide (Rodrigues et al., 2018). There are no other reports currently available so we cannot comment regarding 53BP1, RPA and RAD51.
About the cell cycle of AFS cells, these cells duplicate faster than stem cells from other sources and in fact, AFS cells have a doubling time of 36 hours while ES cells 31-57 hours and iPS cells 48 hours (Pozzobon et al., 2010).
References:
1. Rodrigues, M.; Antonucci, I.; Elabd, S.; Kancherla, S.; Marchisio, M.; Blattner, C.; Stuppia, L. p53 Is Active in Human Amniotic Fluid Stem Cells. Stem Cells Dev 2018, 27, 1507-1517, doi:10.1089/scd.2017.0254.
2. Pozzobon, M.; Ghionzoli, M.; De Coppi, P. ES, iPS, MSC, and AFS cells. Stem cells exploitation for Pediatric Surgery: current research and perspective. Pediatr Surg Int 2010, 26, 3-10, doi:10.1007/s00383-009-2478-8.

Reviewer 2 Report
Dear Colleagues!
The Review is well-written and hardly can be improved in presentation quality yet certain aspects must be addressed by the authors.
Yet the title is a bit "narrower" that coverage of the review and I suggest its change to a broader spectrum.
Furthermore, figure legends provided as parts of figures are now quite informative and I suggest moving them to text.
Fig. 2 is obsolete, this information does not require graphic presentation.
Fig. 3 is misleading as far ar creates a "pluripotency-like" image while Table 1 definitely stipulate multipotency of hAFS. This might be due to unclear explanation of heterogeneity of AFS which contain many different subpopulations and in utero can hardly be described in terms of plasticity correctly.
Finally, the Review needs some "pin-pointed" direction as importance of p53 in tumorogenesis control is widely known.
Do authors consider p53 as an important object for basic research or as a therapeutic target to be investigated in other cell types? What are potential (or existing) strategies for p53 modulation? Overall, some data on p53 expression in freshly isolated AFS is required as far as in vitro data in cultured cells always has some impact of ex vivo existence of stem cells.
Regards, Reviewer
Author Response
Response to Reviewer 2 Comments
The Review is well-written and hardly can be improved in presentation quality yet certain aspects must be addressed by the authors.
Comment 1: Yet the title is a bit "narrower" that coverage of the review and I suggest its change to a broader spectrum.
Response 1: Thank you very much for your suggestion. We suggest the following two titles, whichever title according to the editor is more appropriate for this review.
1. Amniotic fluid cells, Stem cells and p53
2. Amniotic fluid cells, Stem cells and p53: Can we stereotype p53 functions?
Comment 2: Furthermore, figure legends provided as parts of figures are now quite informative and I suggest moving them to text.
Response 2: Thank you for your suggestion. We have shortened the figure legends and wherever required added the information into the text.
Comment 3: Fig. 2 is obsolete, this information does not require graphic presentation.
Response 3: We have removed figure 2 from the manuscript and therefore revised the numbering of the subsequent figures.
Comment 4: Fig. 3 is misleading as far ar creates a "pluripotency-like" image while Table 1 definitely stipulate multipotency of hAFS. This might be due to unclear explanation of heterogeneity of AFS which contain many different subpopulations and in utero can hardly be described in terms of plasticity correctly.
Response 4: Thank you for your comment. We have modified figure 3 to indicate that the AFS cells are not pluripotent but are multipotent cells and in presence of various inducing factors are able to differentiate into all cell types of different lineages. We have made this point clearer by including in the text (refer page 5).
Comment 5: Finally, the Review needs some "pin-pointed" direction as importance of p53 in tumorogenesis control is widely known.
Response 5: Our focus in this review is on AFS cells and the need to investigate the role of p53 in these cells, not just with respect to tumorigenicity but also in cellular processes in normal AFS cells, as this information is currently lacking. Nevertheless, we have added a part in conclusion to clear this point (refer page 9).
Comment 6: Do authors consider p53 as an important object for basic research or as a therapeutic target to be investigated in other cell types?
Response 6: We have added few lines in concluding remarks (refer page 9) in response to your question.
Comment 7: What are potential (or existing) strategies for p53 modulation?
Response 7: We have added a paragraph on the existing strategies for p53 modulation (refer page 7).
Comment 8: Overall, some data on p53 expression in freshly isolated AFS is required as far as in vitro data in cultured cells always has some impact of ex vivo existence of stem cells.
Response 8: We agree with the reviewer that in vitro data in cultured cells has an impact on the ex vivo existence of stem cells. However, all the reports on p53 expression so far have been done only on cultured cells. Therefore, as this is a review we cannot comment on p53 expression in freshly isolated AFS cells as such experimental data is currently not available.

Round 2
Reviewer 2 Report
Dear colleagues!
After revision I have no further comments.
Regarding the title, variant #2 (in my opinion) is preferable.
Regards, Reviewer.